# EFFICIENT LARGE LANGUAGE MODELS FINE-TUNING ON GRAPHS

## ABSTRACT

Learning from Text-Attributed Graphs (TAGs) has attracted significant attention due to its wide range of real-world applications. The rapid evolution of large language models (LLMs) has revolutionized the way we process textual data, which indicates a strong potential to replace shallow text embedding generally used in Graph Neural Networks (GNNs). However, we find that existing LLM approaches that exploit text information in graphs suffer from inferior computation and data efficiency. In this work, we introduce a novel and efficient approach for the end-to-end fine-tuning of Large Language Models (LLMs) on TAGs, named LEADING. The proposed approach maintains computation cost and memory overhead comparable to the graph-less fine-tuning of LLMs. Moreover, it transfers the rick knowledge in LLMs to downstream graph learning tasks effectively with limited labeled data in semi-supervised learning. Its superior computation and data efficiency are demonstrated through comprehensive experiments, offering a promising solution for a wide range of LLMs and graph learning tasks on TAGs.

## 1 INTRODUCTION

Graph neural networks (GNNs) have been widely used for representation learning on graph-structured data (Hamilton, 2020; Ma & Tang, 2021), and they achieve promising state-of-the-art performance on various graph learning tasks, such as node classification, link prediction, and graph classification. Numerous graphs within these domains exhibit nodes that are linked to textual attributes, leading to the prevalence of text-attributed graphs (TAGs). TAGs provide a graph-based framework for representing textual data and illustrating connections between phrases, sentences, or documents through edges. The fusion of textual attributes and graph topology constitutes a valuable wellspring of information, bolstering representation learning in real-world applications such as recommender systems (Jin et al., 2023), citation graphs (Hu et al., 2020; Yang et al., 2016), social networks (Hamilton et al., 2017), and knowledge graphs (Wang et al., 2021).

In the context of GNNs, shallow text embeddings such as Bag-of-Words (Harris, 1954) and Word2Vec (Mikolov et al., 2013) are usually extracted from raw textual data and used as the numerical node attributes in GNNs due to their superior simplicity and efficiency. However, as they do not fully capture the complex textual semantic features, these approaches inherently restrict the performance of downstream tasks. On the other hand, the recursive feature aggregation in GNNs results in the well-known neighborhood explosion problem (Hamilton et al., 2017) such that the computation of each node involves its $L$-hop neighbors with $L$ feature aggregation layers. This not only leads to significant scalability challenges but also limits the exploration of more complex and powerful deep learning techniques such as LLMs for the representation learning on TAGs.

Recently, researchers have begun to explore the potential of pre-trained large language models (LLMs), such as BERT (Devlin et al., 2018), Deberta (He et al., 2020) and DistilBERT (Sanh et al., 2019), for representation learning on TAGs due to their unprecedented capabilities in language understanding and generation across a wide range of tasks. The commonly adopted approach follows a *cascaded structure* (Chen et al., 2023). This entails an initial LLM fine-tuning step on downstream tasks such as node classification. Subsequently, the text embeddings extracted from the fine-tuned LLMs are leveraged as the initial node features for downstream GNNs. Although the cascaded structure is efficient, graph structural information is not incorporated in the fine-tuning of LLMs, resulting in sub-optimal performance. To address this issue, the *iterative structure* has also been

explored for the joint training of LLMs and GNNs. For instance, GLEM (Zhao et al., 2022) trains LLMs and GNNs separately in an iterative manner by generating pseudo labels for each other. In addition, *self-supervised learning* has also been proposed to enhance LLMs by link prediction tasks, exemplified by GIANT (Chien et al., 2021).

The aforementioned works demonstrate the potential of exploiting LLMs on TAGs. However, these approaches still face limitations in *data efficiency* or *computation efficiency*. First, both cascaded and iterative structures encounter significant data inefficiency. When the labeled data is scarce, these methods struggle to effectively transfer the required knowledge for downstream tasks as the fine-tuning strategies do not utilize labeled data efficiently. Second, both iterative structures and the self-supervised learning approach introduce a substantial increase in computational overhead. This elevated computational cost poses scalability challenges, especially when dealing with large-scale datasets. These shortcomings tremendously limit their applications in transferring the rich knowledge of LLMs to facilitate representation learning on TAGs.

In this paper, we aim to develop an efficient LLM fine-tuning algorithm that not only effectively adapts LLMs to downstream tasks with limited labeled data (*data efficiency*) but also exhibits superior scalability (*computation efficiency*). To this end, we first reveal the *encoding redundancy* and *propagation redundancy* in fine-tuning LLMs with GNNs. To reduce these redundancies, we propose a novel LLM-GNN end-to-end training algorithm (LEADING) to efficiently fine-tune LLMs on TAGs. Our empirical study demonstrates that the proposed algorithm exhibits strong scalability comparable to graph-less LLMs fine-tuning. Moreover, it transfers the rich knowledge encoded in LLMs to downstream tasks much more effectively than existing approaches with limited labeled data. Therefore, it offers a promising solution for a wide range of LLMs and graph learning tasks on TAGs.

## 2 RELATED WORK

In this section, we will mainly summarize related works exploring LLMs for learning on TAGs.

**Basic structure of LLMs integrated with GNNs.** To address the limitations posed by the simple cascaded structure, which lacks the ability to harness topological information from the graph, several approaches have recently emerged to enhance Transformer structures or graph representation techniques. Some of these methods incorporate graph structure information into attention computation (Park et al., 2022), while others introduce orthogonal vectors for node and edge tokens to capture structural nuances (Kim et al., 2022). While these enhancements can be effective, they often involve complex attention mechanisms, rendering the direct representation of graph structure a challenging endeavor and significantly increasing the computation complexity of model training.

**Advanced structure of LLMs integrated with GNNs.** To address the aforementioned challenges, researchers have explored approaches that combine Large Language Models (LLMs) with graph-based techniques. Notable examples include Graphformers (Yang et al., 2021), GLEM (Zhao et al., 2022), which employs iterative training as mentioned earlier, and Grad (Mavromatis et al., 2023), which uses GNNs for knowledge distillation on language models. However, these models have their drawbacks. They either rely on a powerful student model to generate high-quality soft labels, which necessitate abundant training data, or introduce significant computational overhead. Additionally, there are other approaches like GIANT (Chien et al., 2021), which uses neighbor prediction to fuse graph into LLMs, and E2EG (Dinh et al., 2022), which incorporates node classification into the joint training process of GIANT. However, these models also face scalability challenges.

**Large-scale GNNs.** A substantial body of existing research is dedicated to enhancing the efficiency and scalability of large-scale GNNs through innovative designs. These designs encompass sampling methods, pre-computing, and post-computing techniques. Sampling methods employ mini-batch training strategies to reduce computation and memory demands by selectively sampling nodes and edges. They mitigate the neighbor explosion issue through practices such as neighbor sampling (Hamilton et al., 2017; Chen et al., 2018a; Zeng et al., 2019) or feature memory updating (Fey et al., 2021; Xue et al., 2023). Pre-computing and post-computing methods separate the feature aggregation and prediction models into distinct stages. Pre-computing involves feature aggregation before training (Wu et al., 2019; Frasca et al., 2020; Sun et al., 2021), while post-computing includes

label propagation after training (Huang et al., 2020). However, these methods have not been shown to be feasible for the end-to-end training or fine-tuning of LLMs.

## 3 METHODOLOGY

GNNs have been proven to be data-efficient due to their excellent prediction performance on semi-supervised graph learning tasks where only very limited labeled data is available. The data efficiency of GNNs can be largely attributed to their ability to integrate node attributes and graph structure information in a unified message-passing framework. Through end-to-end training, it leverages the scarce labeled data to provide informative supervision for the vast pool of unlabeled nodes. However, GNNs' data efficiency comes with the sacrifice of computation efficiency (Hamilton et al., 2017).

From the LLMs perspective, most of the existing approaches exploiting LLMs for learning on TAGs fall short in data efficiency and thus fail to effectively adapt the rich knowledge in LLMs to downstream graph learning tasks as discussed in Section 1 and Section 2. We conjecture that their data inefficiency originates from the fact that existing methods can not fine-tune LLMs with graph learning in an end-to-end manner due to the scalability challenges in both LLMs and GNNs.

Motivated by the above analyses, we aim to improve the data efficiency of fine-tuning LLMs for graph learning on TAGs by developing an end-to-end LLM-GNN training approach. To this end, we have to deal with the scalability challenges coupled with LLMs and GNNs. In this section, we will first analyze the computation redundancy in fine-tuning LLMs with GNNs. Then we propose a novel end-to-end fine-tuning strategy (LEADING) to reduce these redundancies, which leads to a highly efficient and scalable solution. Before that, we first introduce the notations as follows.

**Notations.** A graph is represented by $\mathcal{G} = (\mathcal{V}, \mathcal{E})$ where $\mathcal{V} = \{v_1, \ldots, v_N\}$ is the set of nodes and $\mathcal{E} = \{e_1, \ldots, e_M\}$ is the set of edges. For a text-attributed graph, each node $v_i$ is associated with a sequential of raw text feature. We denote the $d$-dimensional hidden feature vectors of nodes as $\mathbf{X} \in \mathbb{R}^{N \times d}$. The graph structure of $\mathcal{G}$ can be represented by an adjacency matrix $\mathbf{A} \in \mathbb{R}^{N \times N}$, where $\mathbf{A}_{ij} > 0$ when there exists an edge between node $v_i$ and $v_j$, and $\mathbf{A}_{i,j} = 0$ otherwise. The symmetrically normalized graph Laplacian matrix is defined as $\tilde{\mathbf{L}} = \mathbf{I} - \tilde{\mathbf{A}}$ with $\tilde{\mathbf{A}} = \mathbf{D}^{-1/2} \mathbf{A} \mathbf{D}^{-1/2}$ where $\mathbf{D}$ is the degree matrix.

### 3.1 COMPUTATION REDUNDANCY IN LLM-GNN

Although the end-to-end LLM-GNN training potentially offers the advantage of data efficiency, it does come with tremendous scalability limitations. While various sampling approaches have been proposed to improve the scalability and efficiency of GNN training, the integration of LLMs with GNNs in an end-to-end training paradigm introduces its own unique hurdles, primarily due to the huge computation and memory costs of LLMs due to their giant sizes. To address these challenges, we first provide a novel and insightful analysis of computation redundancy in the end-to-end training framework, such as encoding redundancy in LLMs and propagation redundancy in GNNs, which pinpoints the scalability bottleneck we can try to reduce.

**Encoding Redundancy.** In the integration of LLMs with GNNs, we have to adopt mini-batch sampling to reduce the computation and memory costs due to the giant size of LLMs. However, existing sampling strategies of GNNs exhibit heavy redundancy that requires frequently repeated LLM encoding of node features. Taking the mini-batch sampling in Figure 1 as an example, the node features need to be encoded by LLMs multiple times through every epoch, either as target nodes themselves or as neighbors of other target nodes. For example, $V_1$ serves as a target node in Batch 1 and serves as a neighbor node in Batch 2 and Batch 3. However, the LLM embedding of the node features will not have notable changes between the mini-batch iterations due to the nature of model fine-tuning.

The above analysis implies that a significant amount of computation on LLM encoding is redundant. This redundancy becomes particularly considerable when we employ smaller batch sizes, as typically used in LLMs, as well as when we introduce more aggregation layers to capture long-distance information in GNNs. According to our statistical analysis on ogbn-arxiv dataset, during each epoch in the training of a 2-layer GCN with GraphSAGE sampling, the node feature of each node is encoded as a target node only once but as a neighbor node 19 times on average when the batch size

is 1024 (25 times when the batch size is 64). For a 5-layer GCN that requires sampling from 5-hop neighbors, the node feature of each node is encoded 96 times as a neighbor node on average. This statistical analysis clearly verifies the LLM encoding redundancy in mini-batch GNNs.

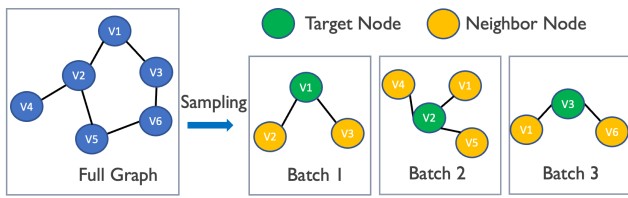

Figure 1: Encoding Redundancy in Mini-batch GNNs.

**Propagation Redundancy.** Besides the LLM encoding redundancy, there also exists propagation redundancy in GNNs. As discovered by a recent work (Xue et al., 2023), the node embedding in the GNN layers will not change notably over the training iterations but the node information is propagated multiple times repeatedly in each iteration to capture long-distance dependency on graphs. This propagation redundancy causes huge sampling, memory, and computation costs that increase significantly with the number of aggregation layers employed.

Next, we will propose a $\underline{L}$arge language models fine-tuning on $\underline{G}$raph (LEADING) algorithm that tackles the encoding redundancy in Section 3.2 and propagation redundancy in Section 3.3.

## 3.2 LEADING: NEIGHBOR DECOUPLING

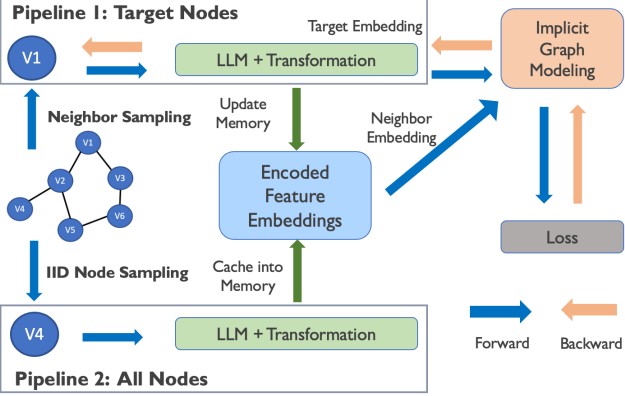

Figure 2: Two-pipeline training process for our proposed LEADING algorithm.

Due to the huge computation and memory cost of LLMs, it is imperative to reduce the redundant LLM encoding computation. Our first observation is that for a sampled subgraph, only the target nodes obtain accurate aggregated features and gradients. On the contrary, the major role of neighbor nodes is to facilitate predictions for target nodes but they may not obtain accurate aggregation features and gradients due to their missing neighbors. In other words, the mini-batch neighbor sampling tries to maximally maintain the neighbors of target nodes but the neighbors of neighbors might be out of the batch. As a result, it is feasible to only use the gradient of target nodes to update the LLMs. The second key observation is that the LLM embedding will not change fast during the fine-tuning stage such that we do not need to update the LLM embedding of neighbor nodes in real time.

**Neighbor Decoupling.** These key observations motivate us to design a novel training algorithm that fully decouples the LLM computation of target nodes and their neighbor node as shown in Figure 2 and Algorithm 1. To reduce the encoding redundancy, we opt to segregate the encoding of target and neighbor nodes into two distinct pipelines. On the one hand, for pipeline 1, the LLM only computes the encoding of target nodes $\mathbf{X}_1$ (line 5). It then retrieves the LLM embedding of neighbor nodes $\mathbf{X}_{\text{neighbor}}$ from the memory bank of encoded feature embedding and concat $\mathbf{X}_1$ and $\mathbf{X}_{\text{neighbor}}$ as the

node embedding of the whole subgraph (line 7) before being fed into GNNs (line 8). On the other hand, for pipeline 2, the LLM randomly sample node features from the whole graph and compute their LLM embedding without requiring gradient (line 3). The computed LLM embedding from both pipeline 1 (line 6) and pipeline 2 (line 4) will be cached in the memory to update the embedding and reduce embedding staleness.

This two-pipeline neighbor decoupling approach offers two significant benefits. First, it completely resolves the encoding redundancy problem, as the encoding times of each node in each epoch are controlled by the batch size used in the second pipeline. For instance, if two pipeline takes the same batch size, each node feature only needs to be encoded by LLM twice, which significantly reduces the computation cost. Second, the memory cost will be significantly reduced since the first pipeline that involves back-propagation training only needs to process target nodes without worrying about the neighbor explosion, which is the key to LLM fine-tuning.

---

**Algorithm 1** LEADING Algorithm

---

**Input:** Input Graph $\mathcal{G} = (\mathcal{V}, \mathcal{E})$, Pre-trained LLM $f(\mathbf{T}, \Theta^0)$
**Output:** Fine-Tuned LLM $f(\mathbf{T}, \Theta^*)$
 1: **Begin**
 2: **for** each mini-batch text $\mathbf{T}_1$ in pipeline 1;
    each mini-batch text $\mathbf{T}_2$ in pipeline 2 **do**
 3:    $\mathbf{X}_2 = f(\mathbf{T}_2, \Theta^k)$: Neighbor Nodes Encoding (without gradients)
 4:    Cache into Memory $\mathbf{M} \leftarrow \mathbf{X}_2$
 5:    $\mathbf{X}_1 = f(\mathbf{T}_1, \Theta^k)$: Target Nodes Encoding (with gradients)
 6:    Cache into Memory $\mathbf{M} \leftarrow \mathbf{X}_1$
 7:    $\mathbf{X}_{\text{in}} = \text{Concat}(\mathbf{X}_1, \mathbf{X}_{\text{neighbor}})$
 8:    $\mathbf{X}_{\text{out}} = \text{GNN}(\mathbf{X}_{\text{in}})$
 9:    Compute Loss and Gradient Update
10: **end for**

---

## 3.3 LEADING: IMPLICIT GRAPH MODELING

While the proposed two-pipeline neighbor decoupling technique can be used in any downstream GNNs, we aim to further improve its scalability by reducing propagation redundancy in GNNs. Motivated by recent advances in implicit models such as Neural ODE (Chen et al., 2018b), IGNN (Gu et al., 2020), and DEQ (Bai et al., 2019), as well as the unified view of graph signal denoising (Ma et al., 2021), we utilize an implicit graph modeling for feature aggregation in just one layer following IGNN and DEQ, which significantly reduces the memory cost of saving intermediate feature embedding. Moreover, motivated by LazyGNN, we reuse the computation results between training iterations to reduce computation costs.

In particular, the forward computation works as follows (similar to APPNP (Klicpera et al., 2018)):

$$\mathbf{X}_{\text{in}}^k = \text{Concat}\{\mathbf{X}_1, \mathbf{X}_{\text{neighbor}}\}, \quad \mathbf{X}_0^k = \mathbf{X}_L^{k-1}, \tag{1}$$

$$\mathbf{X}_{l+1}^k = (1-\alpha)\tilde{\mathbf{A}}\mathbf{X}_l^k + \alpha\mathbf{X}_{\text{in}}^k, \ \forall l = 0, \dots, L-1, \tag{2}$$

where $l$ and $k$ denote the index of layers and training iterations, respectively. The propagation starts from the aggregated feature $\mathbf{X}_L^{k-1}$ from the previous iterations, which reduces propagation cost. The backward propagation works as follows:

$$\mathbf{G}_L^k = \mathbf{G}_0^{k-1}, \quad \mathbf{G}_l^k = (1-\alpha)\tilde{\mathbf{A}}\mathbf{G}_{l+1}^k + \alpha\frac{\partial \mathcal{L}}{\partial \mathbf{X}_L^k}, \ \forall l = L-1, \dots, 0, \tag{3}$$

where $\mathbf{G}_0^k$ provides an approximation for gradient $\frac{\partial \mathcal{L}}{\partial \mathbf{X}_0^k}$. Similarly, the backward propagation starts from the gradient in previous iterations $\mathbf{G}_0^{k-1}$. Finally, the gradient of target nodes can be retrieved from $\mathbf{G}_0^k$ and used for further back-propagation in the LLM $f(\mathbf{T}_1, \Theta^k)$. After the end-to-end fine-tuning, we can utilize the tuned LLM to generate feature embedding, which serves as the initial embedding for any downstream GNNs.

### 3.4 COMPUTATION COMPLEXITY ANALYSIS

**LLM Complexity Analysis.** Suppose $N$ is the total number of nodes and $C$ is the computation complexity of encoding one node feature by LLMs. The total computation complexity of LLM encoding in the proposed LEADING algorithm is $\mathcal{O}(NC)$, which has a nice linear scaling in terms of graph size (number of nodes) but is independent of the graph density (number of edges).

Regarding memory complexity, suppose $\mathcal{O}(S)$ is the memory complexity for executing forward and backward propagation per node. For the mini-batch sampling, suppose $T$ and $B$ are the batch sizes of target nodes and neighbor nodes, respectively. Typically, we have $B \gg T$. Then the total memory complexity for LEADING is $\mathcal{O}(TS)$, which is the same as training LLMs without using graphs. It is much lower than the normal GNN training strategy whose memory complexity is $\mathcal{O}((B+T)S)$. These complexity analyses indicate the intriguing scalability of LEADING in the LLM phase.

**GNN Complexity Analysis.** Suppose $N$ is the total number of nodes, $L$ is the number of propagation layers, and $H$ is the size of hidden units, and $M$ is the number of edges. Performing one feature aggregation in GNNs requires a sparse-dense matrix multiplication, which involves approximately $\mathcal{O}(MH)$ operations. Consequently, the computation complexity for both forward feature aggregations and backward gradient aggregations in GNNs is approximately $\mathcal{O}(2LMH)$ per epoch. It is worth noting that LEADING has fewer layers $L$ compared to existing approaches as described in Section 3.3. This reduction in the number of layers contributes to lowering the overall computation cost in feature aggregation.

For memory complexity, $\mathcal{O}(NH)$ is required to store the intermediate state at each feature aggregation layer, so the total memory complexity for a normal GNN is $\mathcal{O}(LNH)$. However, our algorithm achieves a memory complexity of $\mathcal{O}(NH)$ because we utilize implicit gradient modeling, which does not requires the storage of feature in intermediate layers. Therefore, the memory cost is independent of the number of aggregation layers. This indicates a significant reduction in terms of memory cost.

## 4 EXPERIMENT

In this section, we present experiments to demonstrate the superior data efficiency and computation efficiency of the proposed end-to-end LLM-GNN fine-tuning method, namely LEADING. In particular, we try to answer the following questions: (Q1) Data efficiency: can our LEADING algorithm transfer the knowledge from LLMs to downstream graph learning tasks effectively with limited training data? (Section 4.1) and (Q2) Computation efficiency: can our LEADING algorithm be more scalable compared with other fine-tuning paradigms? (Section 4.2)

**Datasets.** We conduct experiments on both small and large text-attributed graph datasets including Cora (McCallum et al., 2000), PubMed (Sen et al., 2008) and ogbn-arxiv (Hu et al., 2020). We evaluate the effectiveness of LLM fine-tuning by taking semi-supervised node classification problems as the downstream tasks. We randomly split the data into training/val/test sets 10 times for Cora and PubMed and report the mean and variance of accuracy following existing works (Kipf & Welling, 2016). We adopt the default labeling ratios of these datasets, i.e., 20 training nodes per class for Cora and PubMed (low labeling rate) and 53.7% for ogbn-arxiv (high labeling rate).

**Baselines.** We compare the proposed LEADING algorithm with a set of LLM fine-tuning strategies:

- **Shallow Embedding:** Default shallow embeddings provided by PyG (Fey & Lenssen, 2019).

- **Pre-trained LLMs:** LLMs function as simple encoders without fine-tuning on labeled data, and the resulting feature embeddings are used as inputs for downstream GNNs.

- **Supervised-FT LLMs:** LLMs are directly fine-tuned using the labeled data under the supervised setting. Subsequently, the text embedding generated by the fine-tuned LLMs is used as the node embedding for downstream GNNs.

- **GIANT & GLEM:** We choose GIANT (Chien et al., 2021) and GLEM (Zhao et al., 2022) as the major baselines since they exhibit the excellent performance among all existing works. Moreover, GLEM is a representative method of iterative training strategy, while GIANT is a representative method of self-supervised training strategy. It is worth noting that due to the

high training costs associated with GIANT, we only use the pre-trained features provided by their official repository.

**Evaluation setting.** For LLMs, in order to ensure a fair comparison with the baselines across different datasets, we select the same LLMs as used in their respective studies. Specifically, we use BERT (Devlin et al., 2018) as employed in GIANT and DeBERTa (He et al., 2020) as used in GLEM. To evaluate the effectiveness of LLMs fine-tuning, we extract the CLS (classification) embedding from the last hidden states of fine-tuned LLMs as the text embeddings, following the setting in GLEM. For downstream GNNs, we conduct performance comparisons on Cora and Pubmed using two classic GNNs, namely GCN (Kipf & Welling, 2016) and GAT (Veličković et al., 2017). In the case of ogbn-arxiv dataset, we employ GCN and Rev-GAT (Li et al., 2021) following existing works (Chen et al., 2023). We perform all hyperparameter tuning following baselines.

## 4.1 PREDICTION PERFORMANCE

We evaluate the effectiveness of LLM fine-tuning by comparing the prediction accuracy on downstream GNNs. From the accuracy summarized in Table 1, we can make the following observations:

- In the low labeling setting (Cora and Pubmed), LEADING outperforms all other LLM fine-tuning strategies. Notably, compared with Supervised-FT DeBERTa, LEDING significantly boosts the performance of the DeBERTa from $59.2\%$ to $80.6\%$ for GCN and from $57.4\%$ to $81.4\%$ for GAT on Cora. A similar improvement can be observed on PubMed as well.

- On the contrary, GLEM (DeBERTa) performs badly in the low labeling rate setting. These comparisons clearly demonstrate the strong *data efficiency* of the LEADING algorithm since it effectively transfers the knowledge from LLMs with very limited labeled data by end-to-end LLM-GNN training. Note that we found that in the low labeling case, because of the poor quality of generated pseudo labels, GLEM actually achieves its best accuracy in a very special case when the ratio of pseudo labels is set to be 0, which will reduce GLEM to either pre-trained LLM (if using 0 learning rate) or supervised FT LLM. Hence we report the same performance as the best result obtained in pre-trained LLM and supervised FT LLM.

- Another crucial aspect to emphasize is the substantial influence of data efficiency on LLM fine-tuning. In scenarios with limited labeled data, the traditional approach of Supervised Fine-tuning using true labels can negatively impact performance. This observation clarifies instances in our results (Table 1) where the Supervised-FT method achieves similar performance to that of Pre-trained LLMs; this occurrence arises because none fine-tuning is a special case of Supervised-FT when the learning rate is set to 0. This underscores the significance of data efficiency, a key focus of improvement in our LEADING algorithm.

- Comparing Pre-trained DeBERTa with Supervised-FT DeBERTa, the fine-tuning without an end-to-end manner does not provide significant benefits in the low-labeling setting (Cora and PubMed), but it can become more helpful as the amount of training data increases (ogbn-arxiv).

- In the high labeling setting (ogbn-arxiv), LEADING also achieves strong performance. For DeBERTa, LEADING achieves $76.1\%$ and $77.3\%$ accuracy for GCN and Rev-GAT, which are better than GLEM ($75.9\%$ and $76.9\%$), a model that has proven to be very strong in the high labeling setting (Chen et al., 2023). For BERT, LEADING achieves $73.8\%$ and $74.8\%$ accuracy for GCN and Rev-GAT, which are better or comparable with GIANT ($73.3\%$ and $75.9\%$). However, it should be noted that LEADING achieves this remarkable performance with much better computation efficiency and scalability as will be discussed in Section 4.2.

## 4.2 SCALABILITY ANALYSIS

In this section, we investigate the computation efficiency and scalability during the LLM fine-tuning stage. We select BERT and DeBERTa as the LLM architectures since they are used in the baselines of GIANT and GLEM. The results in Table 2 reveal the following noteworthy observations:

- The iterative training strategy such as GLEM and self-supervised training strategy such as GIANT exhibit significantly higher memory cost or running time compared to the cascaded structure such as Supervised-FT. The computational requirements for GIANT are orders of magnitude higher than others, and it runs Out of Memory (OOM) in our experiment.

Table 1: Prediction accuracy (%) of LLM fine-tuning strategies. The **best** are marked as bold.

| | Cora | | Pubmed | | Arxiv | |
|---|---|---|---|---|---|---|
| **METHODS** | **GCN** | **GAT** | **GCN** | **GAT** | **GCN** | **Rev-GAT** |
| Shallow Embedding | **82.0 ± 0.7** | **82.3 ± 0.7** | 78.9 ± 2.0 | 77.7 ± 0.9 | 71.7 | 73.6 |
| Pre-trained DeBERTa | 48.5 ± 1.9 | 51.0 ± 1.2 | 62.1 ± 0.1 | 62.6 ± 0.3 | 45.7 | 47.8 |
| Supervised-FT BERT | — | — | — | — | 73.0 | 73.8 |
| Supervised-FT DeBERTa | 59.2 ± 1.2 | 57.4 ± 2.0 | 62.1 ± 0.1 | 61.6 ± 0.1 | 74.7 | 75.8 |
| GIANT (BERT) | — | — | — | — | 73.3 | 75.9 |
| GLEM (DeBERTa) | 59.2 ± 1.2 | 57.4 ± 2.0 | 62.1 ± 0.1 | 62.6 ± 0.3 | 75.9 | 76.9 |
| LEADING (BERT) | — | — | — | — | 73.8 | 74.8 |
| LEADING (DeBERTa) | 80.6 ± 0.3 | 81.4 ± 0.6 | **79.5 ± 0.8** | **79.3 ± 0.6** | **76.1** | **77.3** |

- Notably, the proposed LEADING achieves a memory cost that is nearly identical to Supervised-FT LLM training without using graphs, which aligns with our expectations. This alignment is attributed to the two-pipeline neighbor decoupling and implicit graph modeling as introduced in Section 3.

- The running time of LEADING is around $0.8$ times higher than that of Supervised-FT, which is reasonable since the two pipelines are run in a sequential manner on the same GPU but it can be easily reduced by parallel computing.

- The memory cost and running time align well with our computational complexity analysis in Section 3.4.

Table 2: Scalability comparison between different LLM fine-tuning strategies.

| **METHODS** | **Memory(GB)** | **Running Time(S)** |
|---|---|---|
| Supervised-FT BERT | 11.5 | 8400 |
| Supervised-FT DeBERTa | 13.6 | 12200 |
| GIANT (BERT) | OOM | N/A |
| GLEM (DeBERTa) | 13.6 | 67634 |
| LEADING (BERT) | 11.7 | 15241 |
| LEADING (DeBERTa) | 13.9 | 22226 |

## 4.3 ABLATION STUDY

**Scalibility study.** We present an ablation study on the memory usage on Cora and the average LLM encoding times of each node on ogbn-arxiv when training BERT with GCN in an end-to-end manner. We assess the memory usage based on two key factors: (1) different batch sizes while keeping 2 hops neighbors (sampling 10 neighbors for the first hop and 5 for the second hop for each node); (2) varying numbers of hops (sampling 10 neighbors for the first hop and 5 for the following hops for each node) while keeping a fixed batch size.

The memory usage in Figure 3 indicates that a significant portion of the computation cost is attributed to the encoding of neighboring nodes. It also demonstrates that LEADING maintains the same memory cost as Supervised-FT LLMs ("Targets Only") and is independent of the number of neighbors included. This highlights a significant scalability advantage. The average LLM encoding times of each node in Figure 4 indicate a considerable level of computational redundancy and this redundancy is affected by both batch size and number of neighbors. The results also show that

our LEADING algorithm significantly reduces this computation redundancy due to the decoupled computation of target nodes and neighbor nodes.

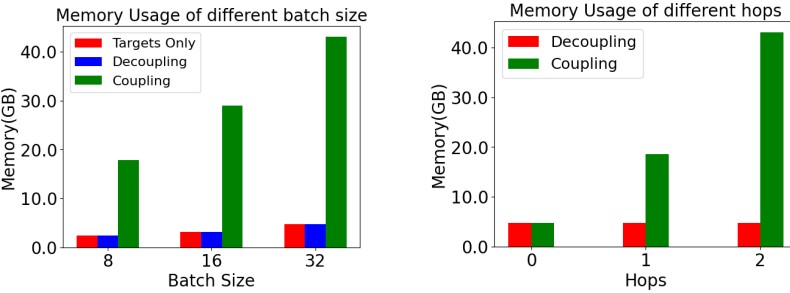

Figure 3: Memory Usage (GB)

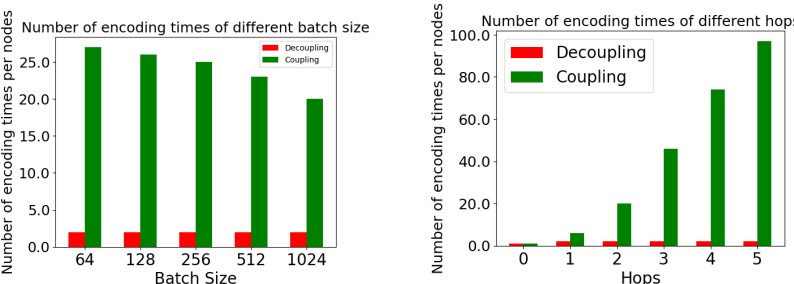

Figure 4: Average LLM Encoding Times

Table 3: Prediction performance comparison between normal LLM-GNN training and LEADING.

| Method / Dataset | Cora | Pubmed |
|---|---|---|
| Normal (Coupling neighbors) | 81.6% | 80.3% |
| LEADING (Decoupling neighbors) | 81.3% | 79.8% |

**Neighbor Decoupling.** We present a simple ablation study to evaluate the impact on prediction performance caused by neighbor decoupling computation. We use DistilBERT (Sanh et al., 2019) with a 2-layer GCN in an end-to-end manner since it is a lightweight version of BERT that can be run faster. We compare two cases: the normal coupling of target and neighbor nodes and our neighbor decoupling approach. The results in Table 3 suggest that neighbor decoupling can achieve closely aligned performance as the coupling method within 0.5% difference, which verifies the rationality of LEADING as discussed in Section 3.2.

## 5 CONCLUSION

Exploring the potential of pre-trained LLMs for representation learning on TAGs has been of significant interest in recent years. However, it comes with significant efficiency issues in the integration of powerful LLMs and GNNs. In this work, we revisit and analyze the limitations of existing approaches with a special focus on data efficiency and computation efficiency. To resolve these limitations, this work develops a novel and efficient LLM-GNN end-to-end fine-tuning algorithm (LEADING) that not only effectively adapts LLMs to downstream graph learning tasks with limited labeled data but also exhibits strong scalability and efficiency. Comprehensive experiments validate its superior prediction performance and efficiency in both low labeling ratio and high labeling ratio settings. The proposed algorithm provides a promising solution for the end-to-end integration of LLMs and GNNs in many impactful real-world applications. In the future, we will explore the integration of the proposed LEADING algorithm with existing Parameter Efficient Fine Tuning (PEFT) approaches for applications of larger scales.

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
