# OpenReview forum: "Efficient Large Language Models Fine-Tuning on Graphs"
_ICLR.cc/2024/Conference — Submitted to ICLR 2024_

### Official Review · Reviewer_k4U4 · 2023-11-01

**Soundness:** 3 good
**Presentation:** 3 good
**Contribution:** 2 fair
**Rating:** 5
**Confidence:** 3

**Summary:**

This paper studies learning from Text-Attributed Graphs (TAGs) using  large language models (LLMs). Specifically, they find that existing LLM approaches that exploit text information in graphs suffer from inferior computation and data efficiency. To address these issues, they propose the LEASUNG fine-tuning algorithm that  not only effectively adapts LLMs to downstream graph learning tasks with limited
labeled data but also exhibits strong scalability and efficiency. Extensive experiments verify the superior performance in terms of both computation and data efficiency.

**Strengths:**

[+] The manuscript is well-presented. The authors clearly present the motivations, the methods and the experiments.

[+] The topic of LLMs for graph is trendy and important in graph learning community.

[+] Extensive experiments are performed to verify the effectiveness in terms of both computation and data efficiency.

**Weaknesses:**

[-] The proposed method can only work for text-attributed graphs and node classification tasks. However, some recent works have validated that LLMs can process nearly all kinds of graphs and node-level/edge-level/graph-level tasks, which makes the contribution of this work being less significant.

[-] The performance of the proposed method is not superior. For example, on Cora dataset, the proposed method equpped with LLMs are inferior to the Shallow Embedding.

[-] The experiments are somewhat limited. Specifically, the authors only evaluate the proposed method in homophilous graphs. I notice that some concurrent works [1,2] in LLMs for graphs report the results on heterophilous graphs. So, how about the performance of the LEADING algorithm perform on heterophilous graphs?

[-] The authors only use some small and out-of-date language models (BERT and DeBERTa). Why not try more powerful LLMs such as GPT-3.5 and LLaMA [3]?

[-] The codes for reproducing the results are not provided.


[1] Exploring the Potential of Large Language Models (LLMs) in Learning on Graphs（arXiv:2307.03393v3）
[2] GRAPHTEXT: GRAPH REASONING IN TEXT SPACE (arXiv:2310.01089v1)
[3] Llama: Open and efficient foundation language models. arXiv preprint arXiv:2302.13971, 2023

**Questions:**

1. How about the performance of the LEADING algorithm perform on heterophilous graphs?

2. Why not try more powerful LLMs such as GPT-3.5 and LLaMA?

---

> ### Author Response · Authors · 2023-11-20
> **Rebuttal by Authors - Post 1**
>
> Dear reviewer,
>
> Thank you for your valuable comments. We appreciate your recognition of our motivation and the experiment design. We have addressed all of your concerns as follows:
>
> **Q1.** The proposed method can only work for text-attributed graphs and node classification tasks. However, some recent works have validated that LLMs can process nearly all kinds of graphs and node-level/edge-level/graph-level tasks, which makes the contribution of this work being less significant.
>
> **A1.** We respectively disagree with this comment. We would like to point out that the proposed training GNN-LLM algorithm is general and can be used for any feature extraction backbones and downstream prediction tasks, enabling data and tasks beyond text-attributed graphs and node classification tasks.
>
> First of all, our proposed method is a general end-to-end GNN training algorithm, which supports any feature extraction backbones that process various kinds of node attributes including but not limited to language, images, video and audio. It works by simply changing the LLM backbone to any feature extraction backbone. However, in this work, we focus on text-attributed graphs
> due to the property of chosen benchmark graph datasets and to ensure a fair comparison with the baselines.
>
> Second, our algorithm framework can support any downstream graph learning tasks. However, exploring LLM for text-attributed graphs is still a pretty new area, and most datasets for graph classification or link prediction do not provide the original raw text attributes. Therefore, we choose node classification tasks following existing state-of-the-art literature such as GIANT [1], GLEM
> [2], and [3] to ensure a fair comparison.
>
> To provide an example of link prediction, we run the proposed algorithm on Cora dataset. We manually partition the links into distinct sets for training, validation, and testing. This division is performed with two distinct split ratios: (1) a low ratio of 10/30/60 for train/valid/test, and (2) a high ratio of 85/5/10 for the same sets. We use GCN as the downstream GNN. To prioritize faster execution and simplicity, we choose DistilBert as the language model. As indicated in the following table, LEADING exhibits performance advantages over the baselines, especially in scenarios with limited training data. This aligns with the findings presented in our paper concerning node classification.
>
> # Link prediction performance AUC (%) on Cora
> | Methods     | Low     | High     |
> |---------|---------|---------|
> | Shallow Embedding  | 74.7  | 94.9  |
> | Pre-trained DistilBERT  | 64.7 | 68.7  |
> | Supervised-FT DistilBERT  | 66.3  | 89.4  |
> | LEADING(DistilBERT)  | 81.8  | 95.2  |
>
> Third, we provide a comprehensive survey of related works in our paper and no existing work can achieve end-to-end fine-tuning of LLMs on graphs. Our proposed algorithm is the first to achieve this goal and shows tremendously improved scalability: the training cost of LEADING is close to the training of LLMs without graphs (refer to the scalability analysis in Table 2 in our submission), which is not thought to be possible in the literature. Our approach significantly minimizes computation redundancy as shown in Fig. 3 and Fig. 4 in our paper.
>
> To summarize, our algorithm is general and novel. It makes significant contributions to the end-to-end training of GNN-LLMs. Please kindly let us know the specific references that use raw text language attributes to complete ”nearly all kinds of graphs and node-level/edge-level/graph-level tasks”, and we will be happy to provide a discussion and comparison.
>
> [1] Chien, Eli, et al. ”Node feature extraction by self-supervised multi-scale neighborhood prediction.” arXiv preprint arXiv:2111.00064 (2021).
>
> [2] Zhao, Jianan, et al. ”Learning on large-scale text-attributed graphs via variational inference.” arXiv preprint arXiv:2210.14709 (2022).
>
> [3] Exploring the Potential of Large Language Models (LLMs) in Learning on Graphs, arXiv:2023.07

---

> ### Author Response · Authors · 2023-11-20
> **Rebuttal by Authors - Continue Post 2**
>
> **Q2.** The performance of the proposed method is not superior. For example, on Cora dataset, the proposed method equipped with LLMs are inferior to the Shallow Embedding.
>
> **A2.** We would like to point out that the proposed LEADING method outperforms all LLM baselines on all datasets. Moreover, LEADING outperforms ”Shallow Embedding” by 4% and around 2% on ogbn-arxiv and PubMed, respectively. We also provide a further experiment based on ogbn-products. For simplicity, we utilize GraphSAGE as the downstream GNN backbone in this context.
>
> In the presented table, LEADING exhibits superior performance compared to all baselines, particularly surpassing shadow embedding by 4.4%. It's noteworthy that GIANT encounters memory issues in our experiments due to high training costs. As a result, we report the accuracy of the well-trained model available from their official repository. These improvements already verify the superiority of our LEADING method.
>
> # Performance and computation cost comparisons on ogbn-products
>
> | Methods     | Accuracy(%)     | Running Time(s)    | Peak Memory Usage(GB)    |
> |---------|---------|---------|---------|
> | Shallow Embedding| 79.7  | ---  | ---  |
> | Pre-trained DeBERTa  | 62.0  | ---  | ---  |
> | Supervised-FT DeBERTa  | 82.2  | 67200 | 25.1  |
> | GIANT | 83.2  | N/A | OOM |
> | GLEM (DeBERTa)  | 83.2  | 356648 | 25.2 |
> | LEADING (DeBERTa)  | 84.1  | 124164 | 25.6 |
>
> For Cora, we believe adding more training data or using a better LLM model can outperform Shallow Embedding. To validate this, we conducted the same experiments using SentenceBERT on Cora and Pubmed with the same low labeling rate. The following table shows it is easy to beat Shallow Embedding, which also verifies the superiority of the proposed LEADING method.
>
> # Performance(%) comparison using SentenceBERT
>
> | Methods     | GCN(Cora)     | GAT(Cora)    | GCN(Pubmed)  | GAT(Pubmed)    |
> |---------|---------|---------|---------|---------|
> | Shallow Embedding  | $82.0 \pm 0.7$ | $82.3 \pm 0.7$  |$78.9 \pm 2.0$  |$77.7 \pm 0.9$  |
> | LEADING (SBERT)   | $\mathbf{83.3 \pm 0.3}$  | $\mathbf{83.8 \pm 0.5}$  |$\mathbf{80.5 \pm 0.4}$  |$\mathbf{80.2 \pm 0.8}$  |
>
> For high labeling rate case, we conduct statistical significance tests on ogbn-arxiv.  We perform 5 times experiments on both our proposed algorithm and the best baselines, GLEM, which has the closest performance as ours on ogbn-arxiv, here we show the results of those performances. The p-value is 9.85e-06 for Rev-GAT and 1.64e-05 for GCN, much less than the commonly used desired significance level 0.01 and highly significance level 0.001. It clearly show that we can reject the null hypothesis. The difference is highly significant.
>
> # Performance(%) on ogbn-arxiv
>
> | Runs | GLEM (GCN)   | LEADING(GCN)     | GLEM (Rev-GAT)      | LEADING(Rev-GAT)   |
> |---------|---------|---------|---------|---------|
> | 1 | 76.90  | 77.42  | 75.90  |76.18  |
> | 2 | 76.80  | 77.30  | 75.95  |76.10 |
> | 3 | 77.00  | 77.26  | 75.93  |76.08  |
> | 4 | 76.85 | 77.34  | 75.86  |76.13 |
> | 5 | 76.78  | 77.34  | 75.87  |76.12 |

---

> ### Author Response · Authors · 2023-11-20
> **Rebuttal by Authors - Continue Post 3**
>
> **Q3.** The experiments are somewhat limited. Specifically, the authors only evaluate the proposed method in homophilous graphs. I notice that some concurrent works [1,2] in LLMs for graphs report the results on heterophilous graphs. So, how about the performance of the LEADING algorithm perform on heterophilous graphs?/How about the performance of the LEADING algorithm
> perform on heterophilous graphs?
>
>
> **A3.** Thanks for your valuable suggestion. Exploring LLM for text-attributed graphs is still a pretty new area, after an extensive search, all benchmark dataset for heterophilic graphs do not provide the original raw text attributes. Therefore, we choose homophilous graphs following existing state-of-the-art literature to ensure a fair comparison and demonstrate our improvements in efficiency and accuracy. For those two cited papers in your comment, GraphText[1] only employs homophilous graphs Cora and Citeseer as TAGs (please refer to the appendix A.2) because it doesn’t require the graph has raw text attributes; and [2] restricts its analysis to some specific nodes which have heterophilous property in homophilous graph Pubmed to draw conclusions. We would be
> pleased to incorporate additional experiments once such graph datasets with raw text attributes become available.
>
> [1] GRAPHTEXT: GRAPH REASONING IN TEXT SPACE (arXiv:2310.01089v1)
>
> [2] Exploring the Potential of Large Language Models (LLMs) in Learning on Graphs（arXiv:2307.03393v3）

---

> ### Author Response · Authors · 2023-11-20
> **Rebuttal by Authors - Continue Post 4**
>
> **Q4.** The authors only use some small and out-of-date language models (BERT and DeBERTa). Why not try more powerful LLMs such as GPT-3.5 and LLaMA?
>
> **A4.** Thanks for this nice suggestion. We would like to point out that the proposed LEADING algorithm is versatile and applicable to any LLM architecture. We choose BERT and DeBERTa as the backbone language models in our initial submission to ensure a fair comparison with state-of-the- art baselines since these are the models they used for evaluation. For instance, we employed BERT
> following GIANT and DeBERTa following GLEM.
>
> Following your suggestion, we are happy to provide further experiments on LEADING using larger models. Based on our current computation resources and the code availability of those larger language models, we choose to fine-tune GPT-2 on Cora and ogbn-arxiv datasets. The results shown in the following table indicate that the proposed LEADING algorithm effectively fine-tunes GPT-
> 2 to achieve better performance, which is consistent with our experiments on other language models. Regarding the computation cost, LEADING is capable of maintaining computational costs nearly identical to supervised fine-tuning of GPT without graphs. The additional running time arises due to the sequential execution of two pipelines in LEADING, yet this can be effectively mitigated through parallel computing. It incurs significantly less computation overhead or memory cost compared to baselines such as GLEM. GIANT runs out of memory in our experiments. It's crucial to emphasize that enhancing model size may not be essential for achieving superior performance, as evident in the comparison between the results of GPT and the DeBERTa presented in our submission. The effectiveness of fine-tuning is influenced by a range of factors beyond mere model size. Note that we found that in the low labeling case because of the poor quality of generated pseudo labels, GLEM actually achieves its best accuracy in a very special case when the ratio of pseudo labels is set to be 0, this will reduce GLEM to supervised FT LLM, hence we report the same performance as supervised FT case.
>
> Besides, Meta requires a request form to access their Llama model parameters. We submitted the request a week ago but have not received a response yet. We will add new results on Llama once we get access to it. We will provide a complete evaluation of these models in our revision.
>
> # LEADING performance(%) with GPT-2
> | Method | GCN(Cora) | GAT(Cora)|GCN(Arxiv)| Rev-GAT(Arxiv)|
> |----------|----------|----------|----------|----------|
> | Pre-trained GPT-2   | 51.9   | 54.7   | 64.8  | 66.9   |
> | Supervised-FT GPT-2   | 70.8   | 71.7   | 73.2   | 73.8   |
> | GLEM(GPT-2)   | 70.8   | 71.7   | 74.0   | 75.1   |
> | LEADING(GPT-2)   | 80.5   | 81.5   | 74.1   | 75.2   |
>
> # Scalability comparison with GPT-2 on ogbn-arxiv
> | Method | Memory(GB) | Running Time(s) |
> |----------|----------|----------|
> | Supervised-FT GPT-2    | 26.8   | 15555   |
> | GIANT   | OOM  | N/A   |
> | GLEM(GPT-2)   | 26.8   | 82930   |
> | LEADING(GPT-2)   | 27.1   | 27920  |
>
> **Q5.** The codes for reproducing the results are not provided.
>
> **A5.** We would like to kindly point out that the code is not mandatory at the submission stage. However, we will be happy to make our code available upon the acceptance of our work. Thanks for your understanding.
>
> To conclude, we believe we have fully addressed all of your concerns. Please kindly let us know if you have any further concerns.

---

> > ### Comment · Reviewer_k4U4 · 2023-11-23
> > **Thanks for the detailed response**
> >
> > Thank you for your detailed response. Based on the following reasons, I have decided to keep my score as 5:
> >
> > 1. Regarding the first point, it is important to note that replacing the LLM backbone would deviate from the original topic of "LLMs for GNNs". The explanation provided seems somewhat forced.
> >
> > 2. Although you have presented new results with GPT2, it is worth mentioning that the original results in Table 1 demonstrate that the proposed method sometimes performs worse than shallow embeddings, especially on the Cora and CiteSeer datasets when considering the standard deviation.
> >
> > Thank you once again for your response!

---

> > > ### Author Response · Authors · 2023-11-23
> > > **Further Response for your Concerns**
> > >
> > > Dear Reviewer,
> > >
> > > Thank you so much for your feedback. We really appreciate that you express your remaining concerns, and we are happy to fully address them.
> > >
> > > (1) We would like to emphasize that our paper only focuses on LLM-GNN architecture for learning on text-attributed graphs, and all of our experiments (including the paper and rebuttal) are based on LLM-GNN. Therefore, our study does not deviate from the original topic of "LLMs for GNNs".
> > >
> > > In our response, we mention the feasibility of replacing the LLM backbone to process other types of attributes, which resolves your concern that our method can only work for text-attributed graphs. In other words, this is just to clarify that your concern is not the weakness of our work. In our work, we focus on LLM since it brings a significant scalability issue. Besides, we believe that if we keep LLM as a feature extractor, it can still take any types of input, not limited to text. For example, GPT-4 can take images as input.
> > >
> > > (2) We believe we have clarified this point in our response **A2**.
> > >
> > > We would like to emphasize that whether the fine-tuned LLMs can outperform Shallow Embedding depends on multiple factors such as the algorithm and the number of training data. Our work focuses on improving the algorithm instead of increasing training data.
> > >
> > > When the training data is sufficient (such as ogbn-arxiv and ogbn-product), we already provide sufficient evidence to show our method significantly outperforms Shallow Embedding.
> > >
> > > For Cora and Pubmed, we only use 20 nodes per class as the training data following the common data split in the literature. With such limited data, it is challenging to train LLMs with so many parameters to outperform Shallow Embedding.  Nevertheless, our algorithm still trains the LLMs much better and more efficiently compared with existing LLM baselines.
> > >
> > > In fact, it is easy to train LLMs to outperform Shallow Embedding by using more training data. To verify this, we set 60% of the nodes as training set, 20% as the validation set, and the remaining 20% as the test set for the Cora dataset. The DeBERTa trained by LEADING clearly outperforms Shallow Embedding, which verifies our conclusions as shown in the following table.
> > >
> > > # Performance(%) on Cora with high labelling rate
> > >
> > > | Method    | GCN     | GAT    |
> > > |---------|---------|---------|
> > > | Shallow Embedding  | $90.9 \pm 2.7$  | $90.6 \pm 3.0$  |
> > > | LEADING (DeBERTa)  | $92.5 \pm 2.3$  | $93.2 \pm 1.8$  |
> > >
> > > Please also check our table "Performance(%) comparison using SentenceBERT" in **A2**.
> > >
> > > Finally, we would like to draw your attention to our scientific contributions in both data efficiency and computation efficiency compared with existing algorithms. Hope our response can fully address your concerns. Please kindly let us know if more evidence or clarification is needed. Thank you.
> > >
> > > Best regards,
> > >
> > > All authors

---

> ### Author Response · Authors · 2023-11-21
> **A Friendly Reminder**
>
> Dear Reviewer k4U4,
>
> We are thankful for your valuable comments and suggestions. We hope that our responses have addressed all your concerns. As there is only one day remaining in the author-reviewer discussion period, we kindly request you let us know if you have any additional questions. We greatly appreciate your further feedback.
>
> If you find our responses satisfactory, we would kindly request an update of the rating to reflect your assessment. Your feedback is valuable to us, and we appreciate your time and consideration.
>
> Best regards,
>
> All authors

---

### Official Review · Reviewer_DmWL · 2023-11-02

**Soundness:** 2 fair
**Presentation:** 3 good
**Contribution:** 2 fair
**Rating:** 5
**Confidence:** 4

**Summary:**

This paper introduces an efficient approach for end-to-end fine-tuning of pre-trained language models (PLMs) on text-attributed graphs (TAGs). The authors argue that existing PLM approaches for TAGs suffer from computation and data efficiency issues, and they propose the LEADING algorithm which maintains computation and memory efficiency similar to graph-less fine-tuning of PLMs. LEADING can effectively transfer rich knowledge to downstream graph learning tasks even with limited labeled data in semi-supervised learning. Experimental results show that LEADING demonstrates superior computation and data efficiency in comparison with existing approaches such as GIANT and GLEM.

**Strengths:**

+ Generalizing the contextualization power of PLMs to structure-rich text data (e.g., text-attributed graphs) is an important and meaningful task. The goal of improving computation and data efficiency in this task is well-motivated.

+ The ideas of removing encoding redundancy and propagation redundancy, as well as the neighbor decoupling strategy, are intuitive and well-explained.

+ The authors conduct experiments on both small and large graphs and perform comprehensive ablation studies, hyperparameter analyses, and scalability studies.

**Weaknesses:**

- This work mainly studies encoder-only PLMs such as BERT and DeBERTa. I do not see how the entire study can be easily generalized to encoder-decoder and decoder-only PLMs from the perspective of either methodologies or evaluation protocols. Therefore, I do not quite agree with the term "Large Language Models" in the title because encoder-only PLMs are usually much smaller than encoder-decoder and decoder-only ones. The study is still a complete one even if only encoder-only PLMs are studied, but the used term somehow overclaims what this study has done.

- Statistical significance tests are missing. It is unclear whether the gaps between LEADING and the baselines are statistically significant or not. In fact, the gaps on arXiv are quite subtle in Table 1, therefore p-values should be reported.

- An important baseline, GraphFormers [1], is cited but not compared.

- The authors only conduct experiments in the semi-supervised node classification task. Besides, all three used datasets are from the academic domain. It is unclear whether the proposed techniques work for other tasks (e.g., link prediction) and other domains (e.g., e-commerce).

[1] Graphformers: Gnn-nested transformers for representation learning on textual graph. NeurIPS'21.

**Questions:**

- Could you conduct statistical significance tests to compare LEADING with the baselines on arXiv?

- Could you report the performance of GraphFormers?

- Could you conduct experiments in other tasks and on graphs from non-academic domains?

---

> ### Author Response · Authors · 2023-11-20
> **Rebuttal by Authors - Post 1**
>
> Dear reviewer,
>
> Thank you for your valuable comments. We appreciate your recognition of our motivation and the proposed idea. We have addressed all of your concerns as follows:
>
> **Q1.** This work mainly studies encoder-only PLMs such as BERT and DeBERTa. I do not see how the entire study can be easily generalized to encoder-decoder and decoder-only PLMs from the perspective of either methodologies or evaluation protocols. Therefore, I do not quite agree with the term ”Large Language Models” in the title because encoder-only PLMs are usually much smaller than encoder-decoder and decoder-only ones. The study is still a complete one even if only encoder-only PLMs are studied, but the used term somehow over-claims what this study has done.
>
> **A1.** Thanks for this nice suggestion. We would like to point out that the proposed LEADING algorithm is versatile and applicable to any LLM architecture. We choose BERT and DeBERTa as the backbone language models in our initial submission to ensure a fair comparison with state-of-the- art baselines since these are the models they used for evaluation. For instance, we employed BERT following GIANT and DeBERTa following GLEM.
>
> Following your suggestion, we are happy to provide further experiments on LEADING using other transformers. We select one of the decoder only language models, GPT-2, conducted similar experiments on the cora and ogbn-arxiv dataset. Due to the utilization of causal (auto-regressive) attention in the decoder-only model, we opt to use the last token for classification instead of the [CLS] token in encoder-only model. One simple example can be found here [1]. The results shown in the following table indicate that the proposed LEADING algorithm effectively fine-tunes GPT-2 to achieve better performance, which is consistent with our experiments on other language models. Regarding the computation cost, LEADING is capable of maintaining computational costs nearly identical to supervised fine-tuning of GPT without graphs. The additional running time arises due to the sequential execution of two pipelines in LEADING, yet this can be effectively mitigated through parallel computing. It incurs significantly less computation overhead or memory cost compared to baselines such as GLEM. GIANT runs out of memory in our experiments. Note that we found that in the low labeling case because of the poor quality of generated pseudo labels, GLEM actually achieves its best accuracy in a very special case when the ratio of pseudo labels is set to be 0, this will reduce GLEM to supervised FT LLM, hence we report the same performance as supervised FT case.
>
> Besides, Meta requires a request form to access their Llama model parameters. We submitted the request a week ago but have not received a response yet. We will add new results on Llama once we get access to it. We will provide a complete evaluation of these models in our revision.
>
> # LEADING performance(%) with GPT-2
> | Method | GCN(Cora) | GAT(Cora)|GCN(Arxiv)| Rev-GAT(Arxiv)|
> |----------|----------|----------|----------|----------|
> | Pre-trained GPT-2   | 51.9   | 54.7   | 64.8  | 66.9   |
> | Supervised-FT GPT-2   | 70.8   | 71.7   | 73.2   | 73.8   |
> | GLEM(GPT-2)   | 70.8   | 71.7   | 74.0   | 75.1   |
> | LEADING(GPT-2)   | 80.5   | 81.5   | 74.1   | 75.2   |
>
> # Scalability comparison with GPT-2 on ogbn-arxiv
> | Method | Memory(GB) | Running Time(s) |
> |----------|----------|----------|
> | Supervised-FT GPT-2    | 26.8   | 15555   |
> | GIANT   | OOM  | N/A   |
> | GLEM(GPT-2)   | 26.8   | 82930   |
> | LEADING(GPT-2)   | 27.1   | 27920  |
>
> [1] https://huggingface.co/docs/transformers/model_doc/gpt2

---

> ### Author Response · Authors · 2023-11-20
> **Rebuttal by Authors - Continue Post 2**
>
> **Q2.** Statistical significance tests are missing. It is unclear whether the gaps between LEADING and the baselines are statistically significant or not. In fact, the gaps on arXiv are quite subtle in Table 1, therefore p-values should be reported./Could you conduct statistical significance tests to compare LEADING with the baselines on arXiv?
>
> **A2.** Thanks for your suggestion. We perform 5 times experiments on both our proposed algorithm and the best baselines, GLEM, which has the closest performance as ours on ogbn-arxiv, here we show the results of those performances. The p-value is 9.85e-06 for Rev-GAT and 1.64e-05 for GCN, much less than the commonly used desired significance level 0.01 and highly significance level 0.001. It clearly show that we can reject the null hypothesis. The difference is highly significant.
>
> # Performance(%) on ogbn-arxiv
>
> | Runs | GLEM (GCN)   | LEADING(GCN)     | GLEM (Rev-GAT)      | LEADING(Rev-GAT)   |
> |---------|---------|---------|---------|---------|
> | 1 | 76.90  | 77.42  | 75.90  |76.18  |
> | 2 | 76.80  | 77.30  | 75.95  |76.10 |
> | 3 | 77.00  | 77.26  | 75.93  |76.08  |
> | 4 | 76.85 | 77.34  | 75.86  |76.13 |
> | 5 | 76.78  | 77.34  | 75.87  |76.12 |
>
> **Q3.** An important baseline, GraphFormers, is cited but not compared./Could you report the performance of GraphFormers?
>
> **A3.** Thanks for your comments. We do not report the performance of GraphFormer due to the following reasons:
>
> (1) GraphFormer faces severe scalability issues since it requires the encoding of all neighboring nodes. This results in significant computation overhead, and we can not run it on large-scale datasets.
>
> (2) The focus of our paper is to provide a flexible LLM fine-tuning framework for arbitrary LLM backbones, but
> Graphformer is a specialized architecture. Therefore, it is hard to make fair and meaningful comparisons.
>
> Due to these reasons, the papers of baselines we compare with do not report the performance of GraphFormer, and we majorly follow their evaluation settings to ensure fair comparisons.

---

> ### Author Response · Authors · 2023-11-20
> **Rebuttal by Authors - Continue Post 3**
>
> **Q4.** The authors only conduct experiments in the semi-supervised node classification task. Besides, all three used datasets are from the academic domain. It is unclear whether the proposed techniques work for other tasks (e.g.,link prediction) and other domains (e.g., e-commerce)./Could you conduct experiments in other tasks and on graphs from non-academic domains?
>
> **A4.**  Thanks for your suggestion. Regarding your first question about the graphs of other tasks, we would like to point out that the proposed training GNN-LLM algorithm is general and can be used for any downstream prediction tasks. However, exploring LLM for text-attributed graphs is still a pretty new area, and most datasets for graph classification or link prediction do not provide the original raw text attributes. Therefore, we choose node classification tasks following existing state-of-the-art literature such as GIANT, GLEM, and [1] to ensure a fair comparison and demonstrate our improvements in efficiency and accuracy.
>
> To provide an example of link prediction, we run the proposed algorithm on Cora dataset. We manually partition the links into distinct sets for training, validation, and testing. This division is performed with two distinct split ratios: (1) a low ratio of 10/30/60 for train/valid/test, and (2) a high ratio of 85/5/10 for the same sets. We use GCN as the downstream GNN. To prioritize faster execution and simplicity, we choose DistilBert as the language model. As indicated in the following table, LEADING exhibits performance advantages over the baselines, especially in scenarios with limited training data. This aligns with the findings presented in our paper concerning node classification.
>
> # Link prediction performance AUC (%) on Cora
> | Methods     | Low     | High     |
> |---------|---------|---------|
> | Shallow Embedding  | 74.7  | 94.9  |
> | Pre-trained DistilBERT  | 64.7 | 68.7  |
> | Supervised-FT DistilBERT  | 66.3  | 89.4  |
> | LEADING(DistilBERT)  | 81.8  | 95.2  |
>
> For your comment about graphs of other domains, we provide a further experiment based on ogbn-products, which is representing an Amazon product co-purchasing network. Nodes represent products sold in Amazon, and edges between two products indicate that the products are purchased together. For simplicity, we utilize GraphSAGE as the downstream GNN backbone in this
> context.
>
> From Table 4, in terms of performance, LEADING outperforms all baselines, including cascaded training (Supervised-FT), self-supervised training (GIANT) and iterative training (GLEM). Regarding computational costs, our findings align with those in our papers: LEADING requires nearly identical memory compared to supervised FT without a graph, and it significantly outperforms GIANT and GLEM. It is worth noting that GIANT runs out of memory in our experiment due to the high training costs associated, so we report the accuracy of the well-trained model provided by their official repository.
>
> # Performance and computation cost comparisons on ogbn-products
>
> | Methods     | Accuracy(%)     | Running Time(s)    | Peak Memory Usage(GB)    |
> |---------|---------|---------|---------|
> | Shallow Embedding| 79.7  | ---  | ---  |
> | Pre-trained DeBERTa  | 62.0  | ---  | ---  |
> | Supervised-FT DeBERTa  | 82.2  | 67200 | 25.1  |
> | GIANT | 83.2  | N/A | OOM |
> | GLEM (DeBERTa)  | 83.2  | 356648 | 25.2 |
> | LEADING (DeBERTa)  | 84.1  | 124164 | 25.6 |
>
> [1] Exploring the Potential of Large Language Models (LLMs) in Learning on Graphs, arXiv:2023.07
>
> To conclude, we believe we have fully addressed all of your concerns. Please kindly let us know if you have any further concerns.

---

> ### Author Response · Authors · 2023-11-21
> **A Friendly Reminder**
>
> Dear Reviewer DmWL,
>
> We are thankful for your valuable comments and suggestions. We hope that our responses have addressed all your concerns. As there is only one day remaining in the author-reviewer discussion period, we kindly request you let us know if you have any additional questions. We greatly appreciate your further feedback.
>
> If you find our responses satisfactory, we would kindly request an update of the rating to reflect your assessment. Your feedback is valuable to us, and we appreciate your time and consideration.
>
> Best regards,
>
> All authors

---

### Official Review · Reviewer_sP5C · 2023-11-04

**Soundness:** 2 fair
**Presentation:** 2 fair
**Contribution:** 3 good
**Rating:** 6
**Confidence:** 4

**Summary:**

This research paper suggests an approach to fine-tuning large language models (LLMs) on text-attributed graphs. The main ideas and contributions are as follows;

- Examining the redundancy, in computing (encoding and propagation) when fine-tuning LLMs using graph networks (GNNs) in an end-to-end manner. This analysis uncovers the scalability limitations.
- Presenting an algorithm called LEADING that reduces computation redundancy by decoupling neighbors and implementing graph modeling. This allows for end-to-end training of LLM GNN to supervised LLM fine-tuning without graphs.
- Demonstrates that LEADING is more resourceful in transferring knowledge from LLMs to downstream graph learning tasks compared to existing methods in scenarios with labeling rates.
- Showing that LEADING achieves scalability and efficiency in LLM fine-tuning and significantly outperforms existing iterative or self-supervised methods that combine LLMs and GNNs.
- Providing complexity analysis and empirical evaluations to highlight the efficiency and scalability benefits of LEADING.

In summary, this paper enables end-to-end training of LLM GNN through techniques that decrease computation redundancy. This facilitates the transfer of knowledge from LLMs into graph learning tasks with both efficiency and scalability, in mind.

**Strengths:**

Importance of the Problem; The paper addresses an issue of tuning Language and Learning Models (LLMs) on graphs, which has implications and challenges, across various domains.

Evaluation of Existing Approaches; The authors conduct an analysis of the limitations in current methods with a particular focus on redundancies in encoding and propagation computations.

Innovative Techniques Proposed; The paper introduces techniques such as neighbor decoupling and implicit graph modeling to overcome the identified limitations. It also presents an approach for training an end to end LLM Graph Neural Network (GNN).

Clear Organization; The paper is well structured and easily comprehensible starting with an introduction followed by a summary of related work.

Clarity and Elaboration; The authors effectively communicate their ideas and techniques using visual aids while providing sufficient algorithmic details.

Experimental Results; Through experiments conducted on datasets the proposed techniques demonstrate advantages, particularly in terms of prediction accuracy when labeled data is limited as well as scalability.

Original Contributions; The analysis of computation redundancies along with the introduced techniques and the end to end training approach are acknowledged as contributions to the field.

Implications; These findings hold implications, for domains that utilize text attributed graphs. They also offer guidance on combining LLMs with GNNs.

**Weaknesses:**

To better understand the proposed techniques and their impact, on data efficiency it would be beneficial to delve into the underlying mechanisms through analysis or intuition.

In order to fully grasp the importance of the techniques it is advisable to conduct ablation studies that elucidate their individual contributions.

To demonstrate the scalability of the algorithm a comprehensive evaluation on a range of graph structures and larger scale datasets would be advantageous.

In the work section it would be beneficial to provide a comprehensive context by discussing existing approaches for efficient training of Graph Neural Networks (GNNs) and implicit models.

Certain parts of the explanation could benefit from in depth details or intuitive explanations especially when describing how the techniques enhance data efficiency.

By expanding the scope of the ablation study and conducting experiments valuable insights, into these proposed methods can be gained.

The conclusion section should be refined to offer a concise yet comprehensive summary of the findings and takeaways.

**Questions:**

1. The analysis findings indicate that there is redundancy, in encoding and propagation. It is not clear how neighbor decoupling and implicit modeling specifically contribute to improved knowledge transfer from LLMs. Could you provide some insight or analysis to explain the underlying mechanisms?

2. The ablation study seems to have limitations. It would be beneficial to conduct ablation experiments to assess the individual contributions of neighbor decoupling and implicit modeling towards the observed improvements.

3. Have you explored graphs or additional datasets beyond those that were tested? Conducting experiments on a larger scale could highlight the scalability benefits more effectively.

4. It would be advantageous to expand upon the work section by discussing approaches such as sampling methods for GNN training and explicit models like Neural ODE/DEQ providing more context and motivation for the techniques employed in your research.

5. Some sections of the paper lack sufficient details. For instance the explanation of how your techniques reduce redundancy can be vague at times. Providing details or intuitive explanations would enhance comprehension.

6. The conclusion feels somewhat abrupt. Please summarize the takeaways and contributions clearly for readers well as discussing potential future directions beyond integration, with PEFT.

7. To further strengthen the paper consider expanding the ablation study incorporating references to work conducting large scale experiments and providing additional intuition and details where necessary.

---

> ### Author Response · Authors · 2023-11-20
> **Rebuttal by Authors - Post 1**
>
> Dear reviewer,
>
> Thank you so much for your positive feedback and valuable comments. It is encouraging to note your recognition of the strengths in our method, including its innovativeness and contributions. We address all of your concerns as follows:
>
> **Q1.** To better understand the proposed techniques and their impact, on data efficiency it would be beneficial to delve into the underlying mechanisms through analysis or intuition. / The analysis findings indicate that there is redundancy, in encoding and propagation. It is not clear how neighbor decoupling and implicit modeling specifically contribute to improved knowledge transfer from
> LLMs. Could you provide some insight or analysis to explain the underlying mechanisms?
>
> **A1.** Thanks for your suggestion. Please allow us to clarify that our algorithm improves knowledge transfer from LLMs to downstream tasks, particularly in scenarios with limited data, resulting in significant data efficiency. This is achieved by supporting end-to-end fine-tuning directly on the downstream tasks, a feature not supported by existing algorithms.
>
> At the beginning of Section 3, we point out that existing algorithms, such as GIANT and GLEM, can not be trained or fine-tuned in an end-to-end manner for the downstream tasks. For instance, GIANT pre-trains the LLMs using self-supervised learning, but their LLMs can not be trained on the downstream tasks due to scalability issues of existing algorithms. Therefore, we believe that due to the mismatch between training and testing, they can not effectively transfer the knowledge of LLMs to downstream tasks, especially when the training data is limited, as shown in our experiments.
>
> Nevertheless, there is no existing work in the literature can support the end-to-end training of LLM-GNN. Therefore, the major motivation of our work is to design an efficient and scalable algorithm to support the end-to-end training of LLM-GNN, which enables the LLM models to be optimized directly for the downstream task, leading to improved knowledge transfer and performance.
>
> Specifically, we first analyze the scalability challenges in end-to-end training of LLM-GNN and identify two primary bottlenecks in Section 3.1: encoding redundancy and propagation redundancy. The proposed techniques including neighbor decoupling and implicit modeling are to solve these two challenges in Section 3.2 and Section 3.3 respectively. The proposed LEADING algorithm is the first to achieve end-to-end training of LLM-GNN, and it shows tremendously improved scalability: the training cost of LEADING is close to the training of LLMs without graphs (refer to the scalability analysis in Table 2 in our submission).

---

> ### Author Response · Authors · 2023-11-20
> **Rebuttal by Authors - Continue Post 2**
>
> **Q2.** In order to fully grasp the importance of the techniques it is advisable to conduct ablation studies that elucidate their individual contributions. / The ablation study seems to have limitations. It would be beneficial to conduct ablation experiments to assess the individual contributions of neighbor decoupling and implicit modeling towards the observed improvements.
>
> **A2.**  Thanks again for your suggestion. As discussed in previous answer, the improvements in prediction accuracy come from the improved scalability to support end-to-end training. Therefore, we would like to provide an ablation study to elucidate the individual contributions of the techniques in terms of algorithm efficiency (running time and memory cost).
>
> We employ the APPNP as GNN backbones and DeBERTa as the language model for our experiments on ogbn-arxiv. We conduct tests in four settings:
> 1. DeBERTa + APPNP (batch size 32)
> 2. DeBERTa + APPNP + Neighbor Decoupling (batch size 32)
> 3. DeBERTa + Implicit Modeling + Neighbor Decoupling (batch size 32)
> 4. DeBERTa + Implicit Modeling + Neighbor Decoupling (batch size 128)
>
> For consistency, we use  two-hop neighbors(5 neighbors for each hop) for all cases. The obtained results reveal that both neighbor
> decoupling and implicit modeling play crucial roles in reducing memory costs. The additional running time is attributed to the sequential execution of two pipelines in our algorithm. This issue can be easily mitigated through parallel training. Moreover, our LEADING algorithm can use a much larger batch size (128) to reduce the running time due to the significantly reduced memory cost.
> This ablation study and analysis elucidates the contributions of our techniques and we will incorporate them into our revision.
>
> # Computation cost comparison on ogbn-arxiv
>
> | Methods    | Running time per epoch(s)     | Peak memory usage(MB)    |
> |---------|---------|---------|
> | DeBERTa+APPNP(BS:32)  | 5770  | 47243  |
> | DeBERTa+APPNP + Neighbor Decoupling(BS:32)  | 12117  | 8522  |
> | DeBERTa+ Neighbor Decoupling + Implicit Modeling(BS:32)  | 12117  | 6259  |
> | DeBERTa+ Neighbor Decoupling + Implicit Modeling(BS:128)  | 2996  | 25103  |

---

> ### Author Response · Authors · 2023-11-20
> **Rebuttal by Authors - Continue Post 3**
>
> **Q3.** To demonstrate the scalability of the algorithm a comprehensive evaluation on a range of graph structures and larger scale datasets would be advantageous./ Have you explored graphs or additional datasets beyond those that were tested? Conducting experiments on a larger scale could highlight the scalability benefits more effectively.
>
> **A3.** Thanks for your suggestion. Here, we present the performance results for ogbn-products, which has over 2.4 M nodes, much larger than ogbn-arxiv. We also provide the associated computational costs. For simplicity, we utilize GraphSAGE as the downstream GNN backbone in this context.
>
> From the table, in terms of performance, LEADING outperforms all baselines, including cascaded training (Supervised-FT), self-supervised training (GIANT) and iterative training (GLEM). Regarding computational costs, our findings align with those in our papers: LEADING requires nearly identical memory compared with supervised FT without a graph, and it significantly outperforms
> GIANT and GLEM. It is worth noting that GIANT runs out of memory in our experiment due to its high training costs, so we report the accuracy of the well-trained model provided by their official repository.
>
> # Performance and computation cost comparisons on ogbn-products
>
> | Methods     | Accuracy(%)     | Running Time(s)    | Peak Memory Usage(GB)    |
> |---------|---------|---------|---------|
> | Shallow Embedding| 79.7  | ---  | ---  |
> | Pre-trained DeBERTa  | 62.0  | ---  | ---  |
> | Supervised-FT DeBERTa  | 82.2  | 67200 | 25.1  |
> | GIANT | 83.2  | N/A | OOM |
> | GLEM (DeBERTa)  | 83.2  | 356648 | 25.2 |
> | LEADING (DeBERTa)  | 84.1  | 124164 | 25.6 |

---

> ### Author Response · Authors · 2023-11-20
> **Rebuttal by Authors - Continue Post 4**
>
> **Q4.** In the work section it would be beneficial to provide a comprehensive context by discussing existing approaches for efficient training of Graph Neural Networks (GNNs) and implicit models./It would be advantageous to expand upon the work section by discussing approaches such as sampling methods for GNN training and explicit models like Neural ODE/DEQ providing more context and motivation for the techniques employed in your research.
>
> **A4.** Thanks for your suggestion. Here we provide a brief introduction about the related work on those efficient training algorithm, and we will add a more comprehensive study to our revision later:
>
> Currently, efficient training strategies for GNNs can be broadly classified into three categories. First, sampling methods adopt mini-batch training strategies to mitigate computation and memory requirements by sampling nodes and edges, mitigating the neighbor explosion issues through techniques like neighbor sampling or feature memory updates. Second, pre-computing or post-computing
> methods separate feature aggregation and prediction models into distinct stages, involving actions such as pre-computing feature aggregation before training or post-processing with label propagation after training. Finally, distributed methods distributes large graphs across multiple servers, parallelizing GNNs training to enhance scalability and efficiency. However, these techniques are not sufficient to support end-to-end GNN-LLM training due to the high computation and memory costs of both. Therefore, existing works choose self-supervised training (GIANT) or iterative training (GLEM), which avoids end-to-end training of LLM and GNNs.
>
> Regarding implicit models, implicit models such as DEQs, Neural ODES and IGNN draw significant research attention due to their memory efficiency. DEQs introduce the concept of deep equilibrium, where the model seeks a fixed-point solution iteratively. It models the forward propagation as solving a fixed point equation. Neural ODEs treat neural networks as continuous dynamical systems, utilizing ordinary differential equations to model the evolution of hidden states. This continuous-depth representation enables seamless integration of depth, offering a flexible and computationally efficient alternative to traditional discrete layer-wise architectures. Both DEQ and Neural ODE can achieve constant memory costs because they do not need to maintain intermediate hidden layers. By using implicit modeling, LEADING also achieves small memory costs, and it can also reduce computation cost by reducing computation redundancy through historical memory.

---

> ### Author Response · Authors · 2023-11-20
> **Rebuttal by Authors - Continue Post 5**
>
> **Q5.** Certain parts of the explanation could benefit from in depth details or intuitive explanations especially when describing how the techniques enhance data efficiency./Some sections of the paper lack sufficient details. For instance, the explanation of how your techniques reduce redundancy can be vague at times. Providing details or intuitive explanations would enhance comprehension.
>
> **A5.** Similar as answer 1, the end-to-end GNN-LMM training enhances the knowledge transfer from LLM to the downstream GNN tasks, which contributes to the improvement of data efficiency. However, no existing works can address the significant computation overhead brought by the end to end training LLM and GNN. We explore the primary challenges arising from encoding redundancy
> and propagation redundancy. Therefore, we propose two techniques, neighbor decoupling and implicit modeling to solve these issues. The primary objective of neighbor decoupling is to address encoding redundancy efficiently. In a normal coupling case, each node serves as a target node once and neighbor nodes many times per epoch (please refer to Fig.1 in our submission), However, our decoupling strategy introduces two distinct pipelines: each node is only encoded as a target node once in pipeline1 and as neighbor nodes once in pipeline2 (please refer to Fig.2), this can significantly reduce redundant encoding times.
>
> Implicit modeling aims to alleviate propagation redundancy. It considers forward propagation as the solution to a fixed-point equation, therefore the feature aggregation can be utilized in just one layer, and the gradient computation involving straightforward first-order derivatives of this equation, there is no need to store intermediate values, in contrast to conventional neural networks. Consequently, the memory complexity remains constant and independent of the number of layers employed. This characteristic represents a notable advantage in reducing memory and computation overhead associated with deep architectures.

---

> ### Author Response · Authors · 2023-11-20
> **Rebuttal by Authors - Continue Post 6**
>
> **Q6.** By expanding the scope of the ablation study and conducting experiments valuable insights, into these proposed methods can be gained./To further strengthen the paper consider expanding the ablation study incorporating references to work conducting large scale experiments and providing additional intuition and details where necessary.
>
> **A6.** Thank you for your valuable suggestion. Regarding the ablation study of the two proposed techniques, we have addressed this in detail in response 2. In the provided table, we illustrate the individual contributions of each technique to our efficiency improvements. The table outlines the impact of each method on the overall enhancement in computation efficiency.
>
> In relation to large datasets, we have conducted additional experiments on ogbn-products, as outlined in response 3. In these experiments, our proposed algorithm demonstrates superior performance compared to all state-of-the-art baselines while maintaining nearly identical computational costs to supervised fine-tuning without graphs. We will incorporate all the results of these experiments
> into our revisions.
>
> **Q7.** The conclusion section should be refined to offer a concise yet comprehensive summary of the findings and takeaways./The conclusion feels somewhat abrupt. Please summarize the takeaways and contributions clearly for readers well as discussing potential future directions beyond integration, with PEFT.
>
> **A7.** Thanks for your suggestion. Following all feedbacks from the reviewers, we will carefully incorporate their comments and suggestions into our work. Then the conclusion section will be thoroughly revised and strengthened after the rebuttal process to provide a clear summary of our contributions and takeaways.
>
> To conclude, we believe we have fully addressed all of your concerns. Thanks again for your support and thoughtful suggestions. Please kindly let us know if you have any further concerns.

---

> ### Author Response · Authors · 2023-11-21
> **A Friendly Reminder**
>
> Dear Reviewer sP5C,
>
> We are thankful for your valuable comments and suggestions. We hope that our responses have addressed all your concerns. As there is only one day remaining in the author-reviewer discussion period, we kindly request you let us know if you have any additional questions. We greatly appreciate your further feedback.
>
> If you find our responses satisfactory, we would kindly request an update of the rating to reflect your assessment. Your feedback is valuable to us, and we appreciate your time and consideration.
>
> Best regards,
>
> All authors

---

### Official Review · Reviewer_LoZg · 2023-11-09

**Soundness:** 2 fair
**Presentation:** 2 fair
**Contribution:** 2 fair
**Rating:** 3
**Confidence:** 4

**Summary:**

This paper discusses

**Strengths:**

1. The idea of combining implicit GNNs and efficient LMs seems quite interesting and intuitive.
2. The summary of the related in this direction is clear and the author identifies the pain points of the current research in the area.

**Weaknesses:**

1. The main claim of this paper is on large language models whereas the experiments are conducted on small language models like "BERT" and "DeBERTa". Given the parameter-efficient tuning, it's not that hard to perform experiments on large models such as GPT-2 and Llama-2.

2. The technical novelty is limited. Caching neighborhoods has been one of the common techniques to speed up GNN-LMs.

3. Evaluations are limited on node classifications only. Given the limited scope and "old" benchmark on node classification, I don't think the contribution of the proposed idea is significant enough.

**Questions:**

1. Can this approach used in other applications such as link prediction and graph classification?

2. What's the performance on larger benchmarks like ogbn-mag/obgn-product, especially the computational cost?

---

> ### Author Response · Authors · 2023-11-20
> **Rebuttal by Authors - Post 1**
>
> Dear reviewer,
>
> We sincerely appreciate your valuable comments. It is encouraging to note your recognition of the combination of implicit GNNs and efficient LLMs in our method as both interesting and intuitive.  We address all of your concerns as follows:
>
>
> **Q1.** The main claim of this paper is on large language models whereas the experiments are conducted on small language models like ”BERT” and ”DeBERTa”. Given the parameter-efficient tuning, it’s not that hard to perform experiments on large models such as GPT-2 and Llama-2.
>
> **A1.** Thanks for this nice suggestion. We would like to point out that the proposed LEADING algorithm is versatile and applicable to any LLM architecture. We choose BERT and DeBERTa as the backbone language models in our initial submission to ensure a fair comparison with state-of-the-art baselines since these are the models they used for evaluation. For instance, we employed BERT following GIANT and DeBERTa following GLEM.
>
> Following your suggestion, we are happy to provide further experiments on LEADING using larger models. We fine-tune GPT-2 on Cora and ogbn-arxiv datasets. The results shown in the following table indicate that the proposed LEADING algorithm effectively fine-tunes GPT-2 to achieve better performance, which is consistent with our experiments on other language models. Regarding the computation cost, LEADING is capable of maintaining computational costs nearly identical to supervised fine-tuning of GPT without graphs. The additional running time arises due to the sequential execution of two pipelines, yet this can be effectively mitigated through parallel computing. It incurs significantly less computation overhead or memory cost compared to baselines such as GLEM. Note that GIANT runs out of memory in our experiments. It's crucial to emphasize that enhancing model size may not be essential for achieving superior performance, as evident in the comparison between the results of GPT and the DeBERTa presented in our submission. The effectiveness of fine-tuning is influenced by a range of factors beyond mere model size.
>
> Besides, Meta requires a request form to access their Llama model parameters. We submitted the request a week ago but have not received a response yet. We will add new results on Llama once we get access to it. We will provide a complete evaluation of these models in our revision.
>
> # LEADING performance(%) with GPT-2
> | Method | GCN(Cora) | GAT(Cora)|GCN(Arxiv)| Rev-GAT(Arxiv)|
> |----------|----------|----------|----------|----------|
> | Pre-trained GPT-2   | 51.9   | 54.7   | 64.8  | 66.9   |
> | Supervised-FT GPT-2   | 70.8   | 71.7   | 73.2   | 73.8   |
> | GLEM(GPT-2)   | 70.8  | 71.7   | 74.0   | 75.1   |
> | LEADING(GPT-2)   | 80.5   | 81.5   | 74.1   | 75.2   |
>
> # Scalability comparison with GPT-2 on ogbn-arxiv
> | Method | Memory(GB) | Running Time(s) |
> |----------|----------|----------|
> | Supervised-FT GPT-2    | 26.8   | 15555   |
> | GIANT   | OOM  | N/A   |
> | GLEM(GPT-2)   | 26.8   | 82930   |
> | LEADING(GPT-2)   | 27.1   | 27920  |

---

> > ### Comment · Reviewer_LoZg · 2023-11-22
> > **Result of baselines with GPT-2 seems to be misleading**
> >
> > Hi authors,
> >
> > Thanks for your additional results using GPT-2. Actually, as a researcher in the field, I am pretty sure FT GPT-2 or GLEM(GPT-2) can produce accuracy higher than 52% on Cora. In my practice, a proper tuned LLM is always better than LMs and in this case I will expect much higher performance than these numbers reported by the authors.

---

> ### Author Response · Authors · 2023-11-20
> **Rebuttal by Authors - Continue Post 2**
>
> **Q2.** The technical novelty is limited. Caching neighborhoods has been one of the common techniques to speed up GNN-LMs.
>
> **A2.** We respectively disagree with this comment. In our paper, we provide a comprehensive survey of related works, and no existing work can achieve end-to-end fine-tuning of LLMs on graphs. Our proposed algorithm is the first to achieve this goal and shows tremendously improved scalability: the training cost of LEADING is close to the training of LLMs without graphs (refer to the scalability analysis in Table 2 in our submission).
>
> Caching neighborhoods is indeed a common technique in accelerating GNNs, and we have cited many related works in our paper but they all require the encoding of both target and neighbor nodes, which are distinctive from our algorithm. Even state-of-the-art scalable GNN training algorithms such as GNNAutoScale (GAS) [1] and GraphFM [2] still encode target nodes and their first-hop neighbors together, which encounter encoding redundancy issues as pointed out in our paper.
>
> In contrast, our proposed algorithm completely decouples target nodes from their neighbors, which is not thought to be possible in the literature. Our approach ensures that in each epoch, each node is only encoded by LLM twice, which minimizes computation redundancy as shown in Fig. 3 and Fig. 4 in our paper. It brings significant cost reduction compared to other memory/cache-
> based GNN training strategies.
>
> Here we also provide a memory cost comparison between LEADING and GAS on ogbn-arxiv dataset, employing the same small batch size of 16. Utilizing DeBERTa as the language model, our results demonstrate distinct advantages. Note that even though GAS only considers 1-hop neighbors, it takes significantly more memory usage.
>
> Our algorithm is novel and makes significant contributions to the end-to-end training of GNN-LMs. Please kindly let us know the specific references that speed up the training or fine-tuning of GNN-LMs, and we will be happy to provide a discussion and comparison.
>
> # Memory cost comparison on ogbn-arxiv
> | Methods    | GAS     | LEADING   |
> |---------|---------|---------|
> | Memory(MB)  | 47780  | 3892  |
>
>
> [1] Fey, Matthias, et al. ”Gnnautoscale: Scalable and expressive graph neural networks via historical embeddings.” International conference on machine learning. PMLR, 2021.
>
> [2] Yu, Haiyang, et al. ”GraphFM: Improving large-scale GNN training via feature momentum.” International Conference on Machine Learning. PMLR, 2022.

---

> ### Author Response · Authors · 2023-11-20
> **Rebuttal by Authors - Continue Post 3**
>
> **Q3.** Evaluations are limited on node classifications only. Given the limited scope and "old" benchmark on node classification, I don't think the contribution of the proposed idea is significant enough. / Can this approach used in other applications such as link prediction and graph classification?
>
> **A3.** We would like to point out that the proposed training LLM-GNN algorithm is general and can be used for any downstream prediction tasks. However, exploring LLM for text-attributed graphs is still a pretty new area, and most datasets for graph classification or link prediction do not provide the original raw text attributes. Therefore, we choose node classification tasks following existing state-of-the-art literature such as GIANT [3], GLEM [4], and [5] to ensure a fair comparison and demonstrate our improvements in efficiency
> and accuracy.
>
> To provide an example of link prediction, we run the proposed algorithm on Cora dataset. We manually partition the links into distinct sets for training, validation, and testing. This division is performed with two distinct split ratios: (1) a low ratio of 10/30/60 for train/valid/test, and (2) a high ratio of 85/5/10 for the same sets. We use GCN as the downstream GNN. To prioritize faster execution and simplicity, we choose DistilBert as the language model. As indicated in the following table, LEADING exhibits performance advantages over the baselines, especially in scenarios with limited training data. This aligns with the findings presented in our paper concerning node classification.
>
> # Link prediction performance AUC (%) on Cora
> | Methods     | Low     | High     |
> |---------|---------|---------|
> | Shallow Embedding  | 74.7  | 94.9  |
> | Pre-trained DistilBERT  | 64.7 | 68.7  |
> | Supervised-FT DistilBERT  | 66.3  | 89.4  |
> | LEADING(DistilBERT)  | 81.8  | 95.2  |
>
>
> [3] Chien, Eli, et al. ”Node feature extraction by self-supervised multi-scale neighborhood prediction.” arXiv preprint arXiv:2111.00064 (2021).
>
> [4] Zhao, Jianan, et al. ”Learning on large-scale text-attributed graphs via variational inference.” arXiv preprint arXiv:2210.14709 (2022).
>
> [5] Exploring the Potential of Large Language Models (LLMs) in Learning on Graphs, arXiv:2023.07

---

> ### Author Response · Authors · 2023-11-20
> **Rebuttal by Authors - Continue Post 4**
>
> **Q4.** What’s the performance on larger benchmarks like ogbn-mag/obgn-product, especially the computational cost?
>
> **A4.** To address your concern, we present the performance results for ogbn-products, where the raw text attributes are available. We also provide the associated computational costs. For simplicity, we utilize GraphSAGE as the downstream GNN backbone in this context.
>
> From the following Table, in terms of performance, LEADING outperforms all baselines, including cascaded training (Supervised-FT), self-supervised training (GIANT), and iterative training (GLEM). Regarding computational costs, our findings align with those in our papers: LEADING requires nearly identical memory compared to supervised FT without a graph, and it significantly outperforms
> GIANT and GLEM. It is worth noting that GIANT runs out of memory in our experiment due to the high training costs associated, so we report the accuracy of the well-trained model provided by their official repository.
>
> # Performance and computation cost comparisons on ogbn-products
>
> | Methods     | Accuracy(%)     | Running Time(s)    | Peak Memory Usage(GB)    |
> |---------|---------|---------|---------|
> | Shallow Embedding| 79.7  | ---  | ---  |
> | Pre-trained DeBERTa  | 62.0  | ---  | ---  |
> | Supervised-FT DeBERTa  | 82.2  | 67200 | 25.1  |
> | GIANT | 83.2  | N/A | OOM |
> | GLEM (DeBERTa)  | 83.2  | 356648 | 25.2 |
> | LEADING (DeBERTa)  | 84.1  | 124164 | 25.6 |
>
> To conclude, we believe we have fully addressed all of your concerns. Please kindly let us know if you have any further concerns.

---

> ### Author Response · Authors · 2023-11-21
> **A Friendly Reminder**
>
> Dear Reviewer LoZg,
>
> We are thankful for your valuable comments and suggestions. We hope that our responses have addressed all your concerns. As there is only one day remaining in the author-reviewer discussion period, we kindly request you let us know if you have any additional questions. We greatly appreciate your further feedback.
>
> If you find our responses satisfactory, we would kindly request an update of the rating to reflect your assessment. Your feedback is valuable to us, and we appreciate your time and consideration.
>
> Best regards,
>
> All authors

---

> ### Author Response · Authors · 2023-11-22
> **Further Response for your Concerns**
>
> Thanks for letting us know your remaining concern about the GPT-2 performance. We are happy to further clarify this.
>
> (1) In our response, we have shown that FT GPT-2 can achieve 70.8% (GCN) and 71.7% (GAT) accuracy on Cora, which aligns with your comment (higher than 52%). Please refer to "Supervised-FT GPT-2 in the Table "LEADING performance (%) with GPT-2" of our response.
>
>
> (2) We want to point out that the performance of GLEM highly depends on the labeling ratio. In all of our experiments (for all LM backbones such as DeBERTa, BERT, and GPT-2), GLEM works pretty well for the cases of high labeling ratios (such as the results for ogbn-arxiv and ogbn-product). However, GLEM does not perform well for cases of low labeling ratios (such as Cora and PubMed).
>
> The underlying reason for the above phenomenon is that GLEM [2] adopts a Two-Stage approach (instead of the end-to-end training as our LEADING algorithm): (1) generate pseudo labels and (2) supervised fine-tuning of LMs on the generated pseudo labels. Therefore, the effectiveness of supervised tuning of LMs on the generated pseudo labels heavily relies on the quality of those pseudo labels. In the common low labeling data split of Cora, the quality of pseudo labels is bad such that fine-tuning of GLEM using these low-quality pseudo labels will even hurt the accuracy.
>
> (3) In our further investigation, we found that in the low labeling ratio case (such as Cora and PubMed), GLEM actually achieves its best accuracy in a very special case when the ratio of pseudo labels is set to be 0. This will reduce GLEM to Supervised Fine-tuning LLMs (only using truth labels), and we missed this special case in our initial experiments. Now we have updated our results for GLEM in all tables. Please refer to our updated results in the response and the updated paper.
>
> (4) We would like to emphasize that our LEADING advances over GLEM from multiple perspectives:
>
> (a) **Data efficiency**: Our end-to-end training algorithm LEADING works better than the two-stage training algorithm GLEM under all cases with both low and high labeling rates, which shows better data efficiency.
>
> (b) **Computation efficiency**: The running time of each single experiment of LEADING is much shorter than GLEM as shown in all of our experiments (please refer to Table 2 in our paper and "Scalability comparison with GPT-2" in our rebuttal).
>
> (c) **Hyperparameters tuning**: LEADING does not require any additional hyperparameters tuning, which further strengthens its efficiency and simplicity. However, the design of GLEM is more complicated and requires additional hyperparameters tuning such as the ratio of generated pseudo labels, the number of iterations of the EM-Step, etc [2][3].
>
>
> Finally, we would like to thank you again for engaging with us in this discussion. We will incorporate all these updated experiments and additional analyses into our revision. We will also make our code publicly available. We hope our clarification can fully address your concerns. Please kindly let us know if you have any further concerns.
>
> [1] Chen, Zhikai, et al. "Exploring the potential of large language models (llms) in learning on graphs." arXiv preprint arXiv:2307.03393 (2023).
>
> [2] Zhao, Jianan, et al. "Learning on large-scale text-attributed graphs via variational inference." arXiv preprint arXiv:2210.14709 (2022).
>
> [3] https://github.com/AndyJZhao/GLEM/tree/main/OGB/ogbn-arxiv

---

### Comment · Area_Chair_y97e · 2023-11-23
**Please check the author response**

Dear Reviewers,

If you haven't done so, please check the author response, as today is the last day of the discussion period. Thank you!

Best,
AC

---

### Meta-Review · Area_Chair_y97e · 2023-12-15

**Metareview:**

This paper proposes an approach combining GNNs with language models for learning on text-attributed graphs. Specifically, the proposed method uses a language model as a feature representation encoder for the textual node features, and then apply GNNs on top of it. The GNN and the language model are then trained/fine-tuned in an end-to-end way. The key innovation is that the proposed method caches some of the neighbor representation calculation, and also stops the gradient to neighbor representations to improve computational efficiency.

Several reviewers commented that the proposed method is relatively straightforward with limited technical contribution. Furthermore, the proposed method isn't performing particularly well in comparison to GNN methods. Given the heavy computation overhead of introducing a language model, the benefit of the proposed method is rather questionable.

**Justification For Why Not Higher Score:**

Simple method that does not have significant performance gain.

**Justification For Why Not Lower Score:**

N/A

---

### Decision · Program_Chairs · 2024-01-16

Reject